# Myeloid cell recruitment versus local proliferation differentiates susceptibility from resistance to filarial infection

Sharon M Campbell[1†], Johanna A Knipper[1], Dominik Ruckerl[1,2], Conor M Finlay[2], Nicola Logan[1], Carlos M Minutti[1], Matthias Mack[3], Stephen J Jenkins[4], Matthew D Taylor[1], Judith E Allen[1,2]*

[1]Centre for Immunity, Infection and Evolution, School of Biological Sciences, University of Edinburgh, Edinburgh, United Kingdom; [2]Wellcome Trust Centre for Cell-Matrix Research, School of Biological Sciences, Faculty of Biology, Medicine & Health, Manchester Academic Health Science Centre, University of Manchester, Manchester, United Kingdom; [3]Department of Internal Medicine II, University Hospital Regensburg, Regensburg, Germany; [4]Centre for Inflammation Research, School of Clinical Sciences, University of Edinburgh, Edinburgh, United Kingdom

*For correspondence:
judi.allen@manchester.ac.uk

Present address: †BioMedicine Design, Pfizer World Wide Research and Development, Dublin, Ireland

Competing interests: The authors declare that no competing interests exist.

**Abstract** Both $T_H2$-dependent helminth killing and suppression of the $T_H2$ effector response have been attributed to macrophages (MΦ) activated by IL-4 (M(IL-4)). To investigate how M(IL-4) contribute to diverse infection outcomes, the MΦ compartment of susceptible BALB/c mice and more resistant C57BL/6 mice was profiled during infection of the pleural cavity with the filarial nematode, *Litomosoides sigmodontis.* C57BL/6 mice exhibited a profoundly expanded resident MΦ (resMΦ) population, which was gradually replenished from the bone marrow in an age-dependent manner. Infection status did not alter the bone-marrow derived contribution to the resMΦ population, confirming local proliferation as the driver of resMΦ expansion. Significantly less resMΦ expansion was observed in the susceptible BALB/c strain, which instead exhibited an influx of monocytes that assumed an immunosuppressive PD-L2+ phenotype. Inhibition of monocyte recruitment enhanced nematode killing. Thus, the balance of monocytic vs. resident M(IL-4) numbers varies between inbred mouse strains and impacts infection outcome.
DOI: https://doi.org/10.7554/eLife.30947.001

## Introduction

*Litomosoides sigmodontis* is a rodent filarial nematode which is used to model the host response to infection with filarial parasites of humans such as *Onchocerca volvulus* and *Wuchereria bancrofti* (*Hoffmann et al., 2000*). Infective L3 stage larvae take 3–6 days to migrate from the skin to the pleural cavity, where they remain for the duration of infection. In susceptible BALB/c mice parasites mature, mate and produce microfilariae that circulate in the bloodstream from ~day 55 post infection (pi). In contrast to BALB/c mice, C57BL/6 mice are considered resistant because the number of adult nematodes recoverable from the pleural cavity declines from ~day 22–55 and parasites do not reach sexual maturity or produce microfilariae (*Hoffmann et al., 2000*; *Graham et al., 2005*). The absence of IL-4, the central cytokine of type two immunity, renders C57BL/6 mice susceptible to *L. sigmodontis* infection, with blood microfilariae detectable at day 60 pi (*Le Goff et al., 2002*).

In response to IL-4Rα stimulation MΦ assume an M(IL-4) activation phenotype characterised by the expression of molecules RELMα, YM1 and arginase-1 (*Stein et al., 1992*; *Doyle et al., 1994*; *Loke et al., 2002*; *Murray et al., 2014*). M(IL-4) have been implicated in nematode killing (*Anthony et al., 2006*; *Zhao et al., 2008*; *Esser-von Bieren et al., 2013*; *Bonne-Année et al.,*

2013) but paradoxically also in suppression of the $T_H2$ immune response (*Nair et al., 2009*; *Pesce et al., 2009b*; *Pesce et al., 2009a*). We have previously reported that IL-4 induces the proliferative expansion of F4/80[hi] resident MΦ (resMΦ) in the pleural cavity during *L. sigmodontis* infection, with minimal blood monocyte recruitment (*Jenkins et al., 2011*; *Jenkins et al., 2013*). F4/80[hi] resMΦ of the serous cavities are initially derived from F4/80[hi] yolk-sac and foetal liver MΦ, prior to the establishment of haematopoietic stem cells (HSCs) which give rise to F4/80[lo] bone marrow derived MΦ (bmMΦ) (*Yona et al., 2013*; *Schulz et al., 2012*; *Ginhoux et al., 2010*). F4/80[hi] resMΦ and recently recruited F4/80[lo] bmMΦ possess distinct M(IL-4) activation profiles upon stimulation with IL-4 (*Gundra et al., 2014*).

MΦ are one of the most abundant cell populations within the pleural cavity during *L. sigmodontis* infection, yet the composition of the myeloid compartment over the course of infection in resistant and susceptible strains remains unexplored. Consequently, we decided to compare the dynamics of MΦ accumulation during *L. sigmodontis* infection between C57BL/6 and BALB/c mice. We specifically asked whether differences in MΦ origin, accumulation and activation phenotype correlate with functional consequences regarding parasite clearance and whether these differences could resolve dichotomous functions associated with M(IL-4). We demonstrate striking differences in myeloid cell dynamics between resistant C57BL/6 mice and susceptible BALB/c mice. In particular, the F4/80[hi] resMΦ population in both naïve and infected C57BL/6 mice was steadily replenished by bmMΦ that assume residency markers GATA6 and CD102. Infection of C57BL/6 mice led to proliferative expansion of the F4/80[hi] resMΦ population, regardless of origin. In contrast, in BALB/c mice, recently recruited bmMΦ failed to successfully integrate into the resident niche and assumed an PD-L2[+] M(IL-4) phenotype that contribute to host susceptibility.

## Results

### Resistant C57BL/6 mice show enhanced F4/80[hi] MΦ accumulation

Our first objective was to compare the cellular infiltrate in resistant C57BL/6 versus susceptible BALB/c strains across the infection time course. We therefore examined the pleural compartment of infected animals at day 11, 28, 35 and 50 pi. These time points reflected known F4/80[hi] MΦ proliferation (*Jenkins et al., 2011*), L4-adult worm moulting (*Hoffmann et al., 2000*), the peak of cellular accumulation (*Babayan et al., 2003*) and a final time point prior to microfilarial development (*Hoffmann et al., 2000*), respectively. As early as day 11 pi there were significantly more cells isolated from the pleural cavity of resistant C57BL/6 mice as compared to susceptible BALB/c mice (*Figure 1A*). This greater cellularity was due predominantly to significantly higher MΦ numbers along with more B cells (*Figure 1A*). The remaining infiltrate consisted of neutrophils, T cells and eosinophils (*Figure 1—figure supplement 1*). Neutrophil numbers were significantly higher in infected C57BL/6 mice at day 28 and 35 pi but represented less than 1% of the total cells. Only eosinophils and T cells were significantly higher in BALB/c mice and only at day 50, a difference likely due to the presence of parasites in BALB/c but not C57BL/6 mice at the later time point. We next assessed the contribution of F4/80[hi] and F4/80[lo] MΦ to the increase in total MΦ number. Due to the emergence of a Ly6C[+] population within the pleural cavity of infected mice, cells expressing lower levels of F4/80 were further subdivided into F4/80[lo]MHC[hi]Ly6C[-] monocyte-derived macrophages (F4/80[lo]) and recently recruited F4/80[lo]MHC[lo-hi]Ly6C[+] monocytes (*Figure 1B*). Thus, the monocyte gate contained Lin[-](CD19, Ly6G, SiglecF, TCRβ), CSF-1R[+], F4/80[lo], Ly6C[+] cells. This gating strategy revealed that the enhanced MΦ number observed in C57BL/6 mice as compared to BALB/c mice was reflective of an expanded F4/80[hi] population (*Figure 1C*). There were also more F4/80[lo] MΦ in the C57BL/6 strain but only at day 11 pi. F4/80[lo] MΦ numbers did not differ significantly between strains throughout the remainder of the time course (*Figure 1C*). Notably, although C57BL/6 mice displayed increased numbers of monocytes throughout the infection as compared to their naïve controls, BALB/c mice were marked by a significant influx of Ly6C[+] monocytes first observed at day 35 pi (*Figure 1C*). Thus, the immune response to the parasite in C57BL/6 mice was characterised by increased numbers of F4/80[hi] MΦ and B cells from day 11 pi. In contrast, susceptible BALB/c mice showed significantly less F4/80[hi] cell expansion and an influx of Ly6C[+] monocytes from day 35 pi.

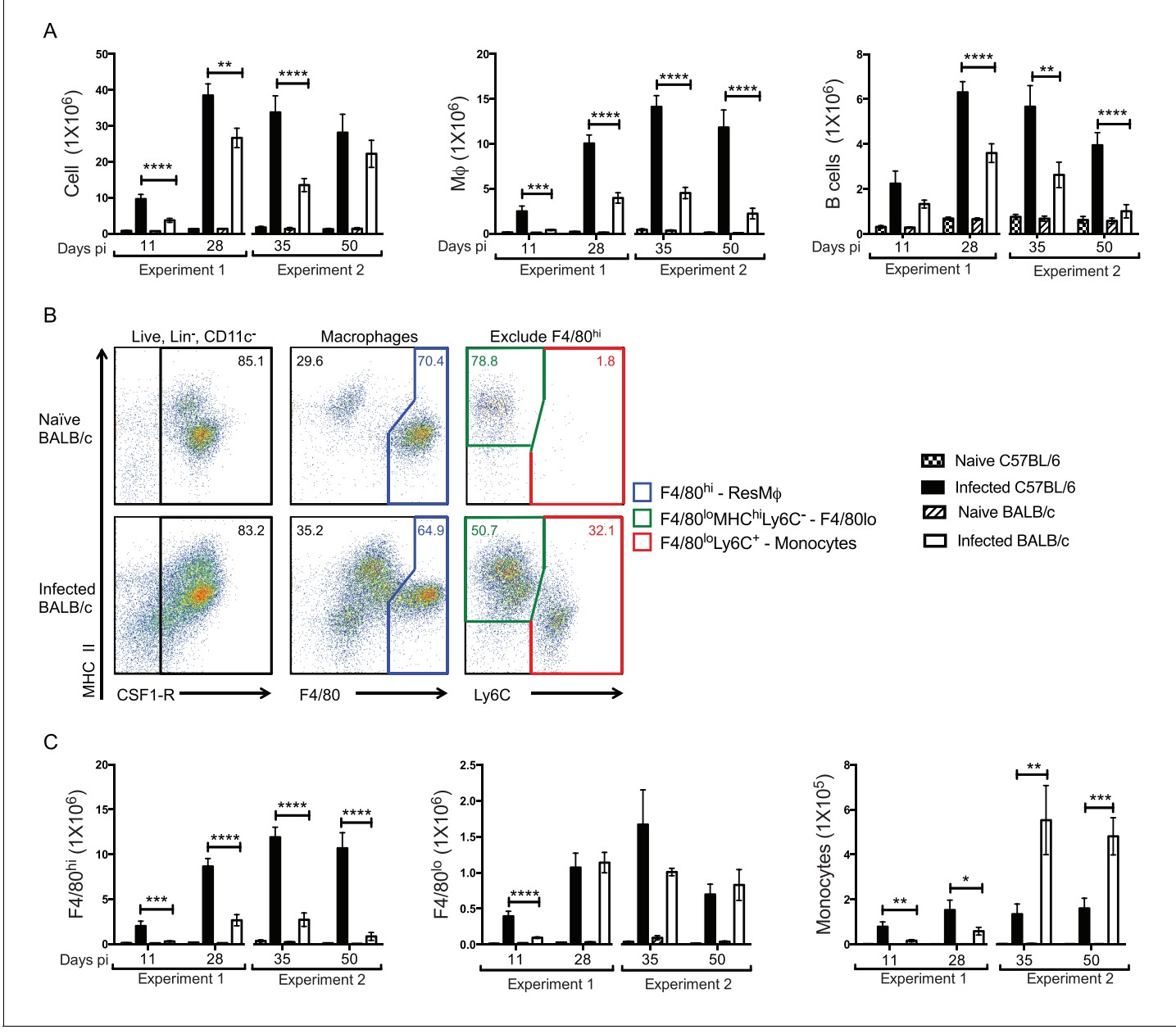

**Figure 1.** Enhanced F4/80[hi] MΦ and B cell numbers are associated with resistance. (**A**) Difference in total exudate cell, MΦ and B cell number between naïve and *L. sigmodontis* infected C57BL/6 and BALB/c mice at day 11, 28, 35 and 50 pi. MΦ were identified as live, Lin[−] (CD19, Ly6G, SiglecF, TCRβ) CSF1R[+] CD11c[−]. (**B**) Representative plots from naïve and infected BALB/c mice at d35 pi, demonstrating the gating strategy used to identify and divide the MΦ population based on expression of F4/80, MHC and Ly6C. (**C**) Number of F4/80[hi], F4/80[lo] and monocytes isolated from pleural cavity of mice in (**A**). Presented are the data from two separate time course experiments (day 11 and 28 and day 35 and 50), each of which is representative of three independent experiments with 6 mice/group/time point. *p<0.05, **p<0.01, ***p<0.0001, ****p<0.00001 as determined by a 2-way ANOVA comparing infected C57BL/6 with infected BALB/c mice at each time point. Error bars represent the mean ± SEM.

DOI: https://doi.org/10.7554/eLife.30947.002

The following figure supplement is available for figure 1:

**Figure supplement 1.** Breakdown of cell populations in the pleural exudate.

DOI: https://doi.org/10.7554/eLife.30947.003

## Susceptible BALB/c mice fail to maintain the F4/80<sup>hi</sup> resMΦ population

To further characterise the MΦ dynamics in the pleural cavity, we profiled the percentage contribution of F4/80$^{hi}$, F4/80$^{lo}$ and monocyte populations to the MΦ compartment as a whole within each strain of naïve and infected animals (*Figure 2A*). In naïve C57BL/6 control mice the F4/80$^{hi}$ population constituted 80–90% of the total MΦ population from 8 to 15 weeks of age (*Figure 2B*). The remaining 10–20% of the MΦ compartment was composed of F4/80$^{lo}$ MΦ, with almost negligible contribution of monocytes in the C57BL/6 strain (*Figure 2B*). Interestingly, the proportion of F4/80$^{hi}$ cells contributing to the MΦ pool in naïve C57BL/6 mice declined slightly with age (87 ± 3% to 73 ± 3%) and infection prevented this age-related decline, maintaining the F4/80$^{hi}$ population at ~90% of the total MΦ pool (*Figure 2B*).

The transcription factor GATA6 and cell surface protein CD102 have been identified as markers of residency expressed by the F4/80$^{hi}$ MΦ population within the peritoneal and pleural spaces of C57BL/6 mice (*Okabe and Medzhitov, 2014*; *Rosas et al., 2014*; *Bain et al., 2016*). Consistent with these reports the F4/80$^{hi}$ population in C57BL/6 mice was positive for GATA6 and CD102 (*Figure 2—figure supplement 1*) which was reflected in ~80–90% of the total MΦ pool being positive for GATA6 and CD102 at 8 weeks of age or d11 pi (*Figure 2C*). In naïve C57BL/6 mice, the percentage of MΦ expressing GATA6 was significantly reduced by 15 weeks of age (*Figure 2C*) reflective of the age-related decline in the proportion of F4/80$^{hi}$ cells (*Figure 2B*). In contrast, there was no decline in GATA6 within the MΦ compartment of infected C57BL/6 mice and CD102 expression was sustained at a higher level in infected mice (*Figure 2C*), consistent with maintenance of the F4/80$^{hi}$ phenotype at 90% of the total MΦ pool (*Figure 2B*).

The MΦ dynamics of the BALB/c strain were distinct from the outset, with the F4/80$^{hi}$ population representing 70–80% of the MΦ compartment at 8 weeks of age/day 11 pi, slightly lower than what was observed in the C57BL/6 strain (*Figure 2D*). A more pronounced decline in the proportion of F4/80$^{hi}$ cells contributing to the total MΦ compartment was observed in both naïve and infected BALB/c mice over the time course. This decline was marked by a corresponding increase in the percentage of F4/80$^{lo}$ MΦ in the cavity (*Figure 2D*). In contrast to C57BL/6 mice, *L. sigmodontis* infection in the BALB/c strain did not prevent the age-related decrease in the proportion of F4/80$^{hi}$ cells. In addition, infection induced recruitment of Ly6C$^{+}$ monocytes from the bone marrow, which further reduced the relative contribution of F4/80$^{hi}$ cells. As a result, by day 50 pi in the BALB/c strain, F4/80$^{lo}$ MΦ and monocytes represent >50% of the myeloid pool (*Figure 2D*). Consequently, the proportion of myeloid cells expressing residency markers GATA6 and CD102 declined significantly over the time course in both naïve and infected BALB/c mice (*Figure 2E*).

Together these data suggest that the proportion of resident F4/80$^{hi}$ MΦ within the pleural space declines in an age dependent manner and that this decline is more dramatic in BALB/c mice than C57BL/6 mice. Further, infection of C57BL/6 mice but not BALB/c mice was able to prevent the age-related decline of the resident compartment. The stark differences between the two strains in both naïve animals and at day 35 pi was highlighted using dimensionality reduction analysis of multiparametric flow cytometry in combination with traditional population gating (*Figure 2—figure supplement 2*).

## Resistant C57BL/6 mice maintain the F4/80<sup>hi</sup> resMΦ population through proliferative expansion

We have previously demonstrated a peak of F4/80$^{hi}$ proliferation in the pleural cavity at day 10 pi with *L. sigmodontis* (*Jenkins et al., 2013*). To address whether the enhanced F4/80$^{hi}$ resMΦ numbers at day 35 (*Figure 1A*) is a result of continued proliferation after d10 pi, we assessed Ki67 expression levels over the infection time course. Cells that exhibit a high level of Ki67 expression after staining with the BD clone B56 have been shown to be actively dividing (*Jenkins et al., 2013*; *Davies et al., 2011*). Of the time points measured, F4/80$^{hi}$ resMΦ proliferation peaked above naïve levels at day 11 pi only (*Figure 3A*) and was significantly greater in the C57BL/6 strain compared to BALB/c (*Figure 3—figure supplement 1A*). Although we only observed Ki67$^{hi}$ staining above background at day 11 (*Figure 3A*), the F4/80$^{hi}$ MΦ population continued to increase between day 28 and day 35 (*Figure 1*). Experiments in *Ccr2$^{-/-}$* C57BL/6 mice demonstrated that F4/80$^{hi}$ MΦ expansion at d28 pi was not significantly altered making it unlikely that bone marrow derived cells were contributing to the increased numbers (*Figure 3—figure supplement 1B and C*). We thus

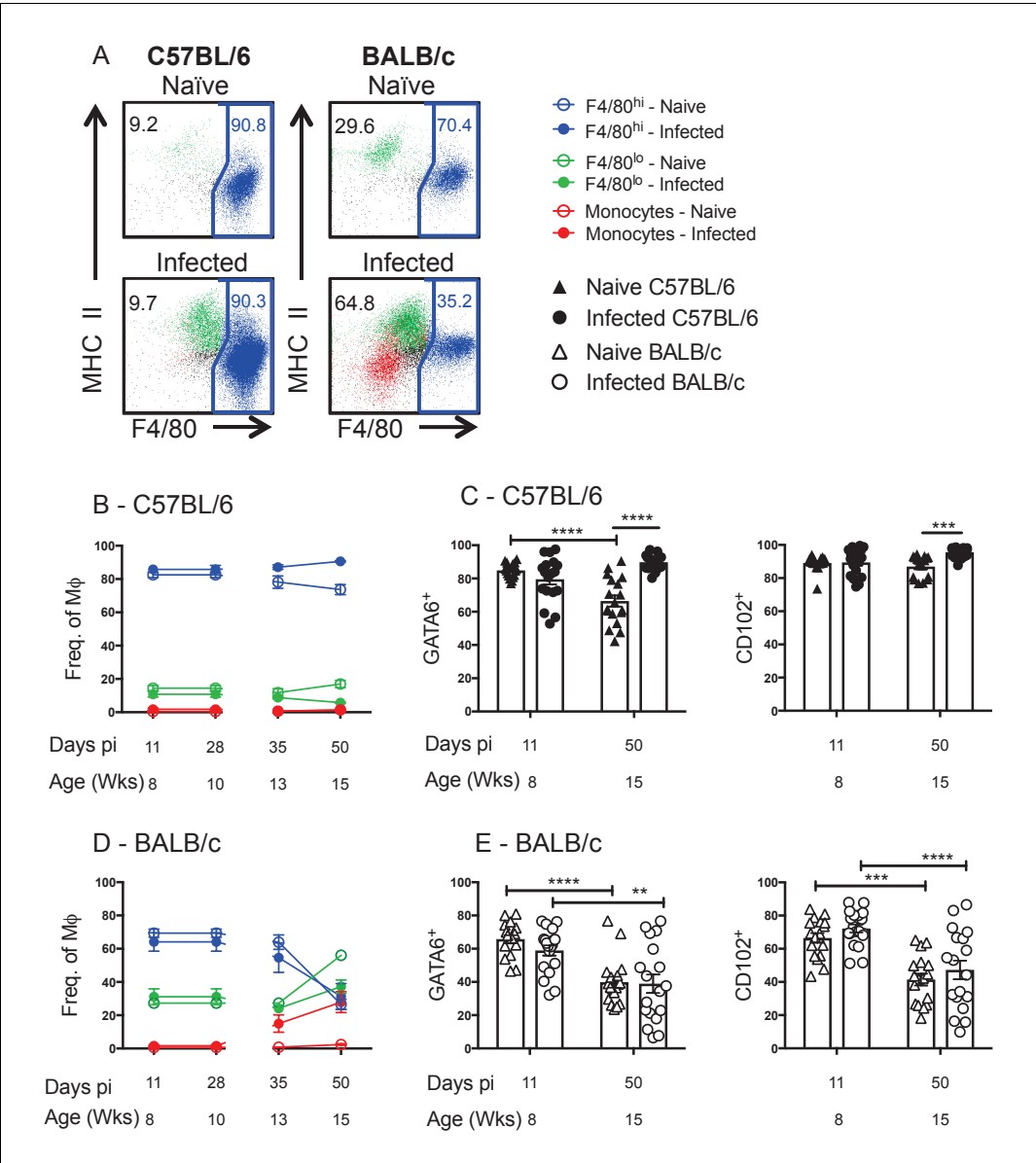

**Figure 2.** Residency is maintained in resistant C57BL/6 mice and lost in susceptible BALB/c mice. (**A**) Representative FACS plots of MΦ subpopulations from the pleural cavity of naïve and d35 pi C57BL/6 or BALB/c mice. Blue: F4/80$^{hi}$, Green: F4/80$^{lo}$, Red: monocytes. Percentage of F4/80$^{hi}$, F4/80$^{lo}$ and monocytes contributing to the MΦ compartment as a whole in (**B**) C57BL/6 and (**D**) BALB/c mice. MΦ expression of GATA6 and CD102 in naïve and *L. sigmodontis* infected (**C**) C57BL/6 and (**E**) BALB/c mice. (**A, B, D**) Presented are the data from two separate time course experiments (day 11 and 28 and day 35 and 50), each of which is representative of three independent experiments with 6 mice/group/time point. (**C** and **E**) Presented are the pooled data from three independent experiments. **p<0.01, ***p<0.0001, ****p<0.00001 as determined by a 2-way ANOVA on each time point. Error bars represent the mean ± SEM.

DOI: https://doi.org/10.7554/eLife.30947.004

The following figure supplements are available for figure 2:

**Figure supplement 1.** GATA6 and CD102 expression on pleural myeloid cells.

DOI: https://doi.org/10.7554/eLife.30947.005

**Figure supplement 2.** Phenotyping of pleural cavity myeloid cells illustrates differences between BALB/c and C57BL/6 macrophage populations during *L. sigmodontis* infection.

DOI: https://doi.org/10.7554/eLife.30947.006

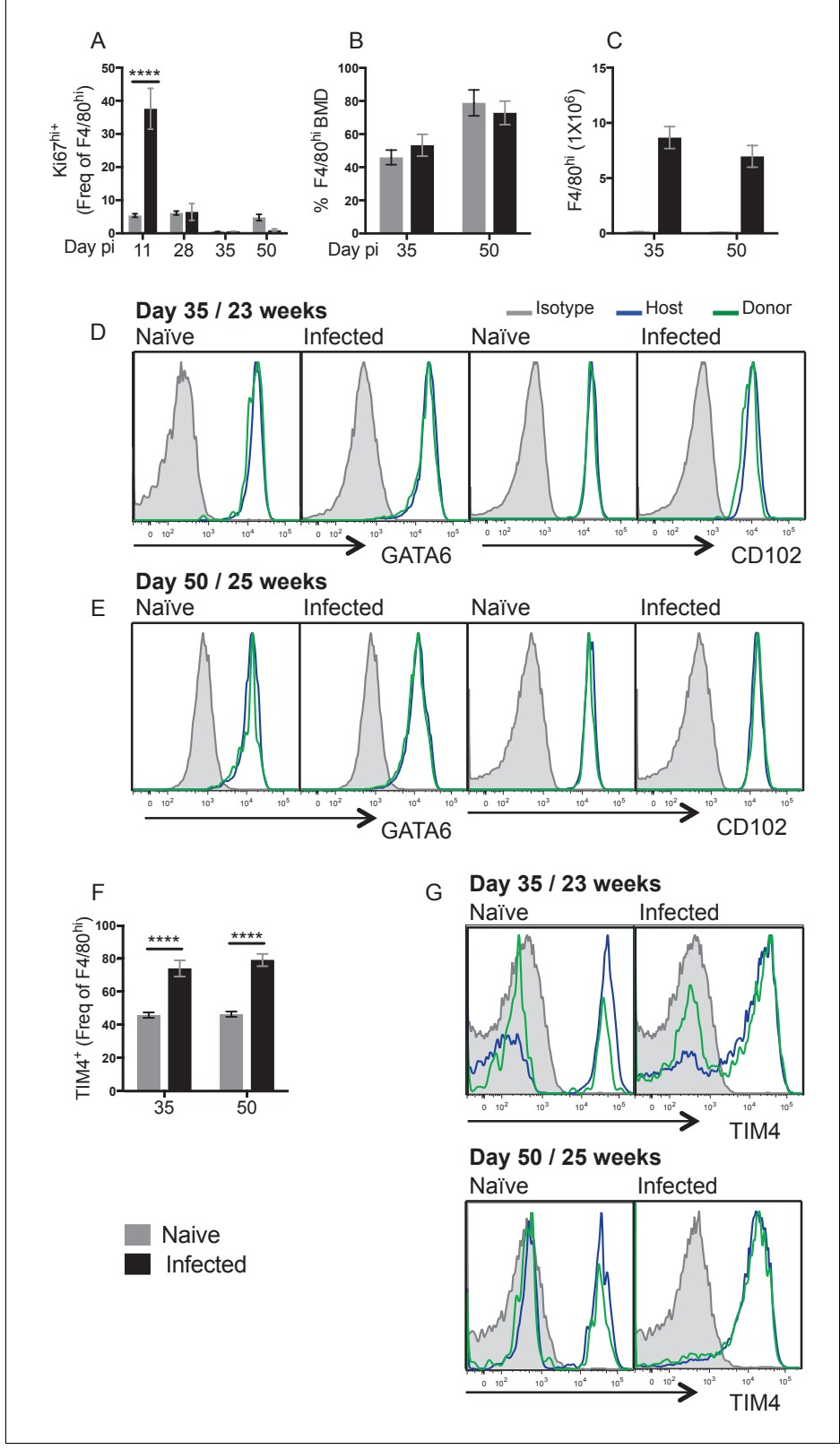

**Figure 3.** Local proliferation accounts for enhanced F4/80<sup>hi</sup> cell number in resistant C57BL/6 mice. (**A**) Expression of high levels of Ki67 by pleural F4/80<sup>hi</sup> MΦ from naïve (grey bars) or infected (black bars) C57BL/6 mice at d11, d28, d35 and d50 pi. (**B**) Percentage of Bone Marrow Derived (BMD) cells contributing to F4/80<sup>hi</sup> population at d35 and d50 pi in naïve and *L. sigmodontis* infected partial bone marrow chimeric C57BL/6 mice. (**C**) F4/80<sup>hi</sup> cell

*Figure 3 continued on next page*

*Figure 3 continued*

number at d35 and d50 pi from animals in (**B**). (**D** and **E**) Expression of GATA6 and CD102 by host/donor derived F4/80[hi] MΦ at d35 and d50 pi (**F**) expression of TIM4 by F4/80[hi] MΦ at d35 and d50 pi in naïve and *L. sigmodontis* infected partial bone marrow chimeric C57BL/6 mice (**G**) Expression of TIM4 at d35 and d50 pi. Data in (**A**) is the representative of 3 experiments with 6 mice per group, data in (**B–D**) are representative of two independent experiments with 10 mice/group/time point. \*\*\*p<0.0001 as determined by a 2-way ANOVA on each time point. Error bars represent the mean ± SEM.

DOI: https://doi.org/10.7554/eLife.30947.007

The following figure supplements are available for figure 3:

**Figure supplement 1.** Absence of *Ccr2* does not affect F4/80hi MΦ accumulation at day 28 pi.

DOI: https://doi.org/10.7554/eLife.30947.008

**Figure supplement 2.** Schematic of experimental procedure used to generate partial-bone marrow chimeric mice on the C57BL/6 background.

DOI: https://doi.org/10.7554/eLife.30947.009

hypothesised that additional proliferative bursts between day 11–35 pi would account for this, perhaps in combination with reduced cell death. Measuring proliferation every day would be challenging due to infective larval availability, we therefore decided to use shielded bone marrow chimeras to firmly establish whether the increased F4/80[hi] population at day 35 and 50 pi was the result of local F4/80[hi] expansion or recruitment and conversion of monocytes into the F4/80[hi] pool. Shielding the upper body including the pleural space of CD45.1[+/+] C57BL/6 mice from radiation, allows the generation of chimeric mice in which the degree of bone marrow contribution to cells in protected tissues can be determined (*Jenkins et al., 2011*; *Murphy et al., 2008*; *Bain et al., 2016*; *Baratin et al., 2017*). The proportion of a particular cell population that is bone marrow derived can be calculated by dividing the percentage of donor CD45.2[+] cells in the tissue of interest with that observed in blood Ly6C[hi] monocytes. Full bone marrow replacement will show equivalent chimerism with the blood (*Figure 3—figure supplement 2*). Because of the need to recover from irradiation, mice were on average 9 weeks older than those in the experiments described above.

We were initially surprised to find that ~50% of the F4/80[hi] population was derived from the bone marrow by 23 weeks of age in naïve and infected animals at day 35 pi. Over the next two weeks chimerism increased further to ~80% (*Figure 3B*). Critically, despite near identical chimerism between infected and naïve mice at both day 35 and day 50, infected mice possessed a MΦ population size that was 27-fold larger (*Figure 3C*). Thus the increased contribution of bmMΦ to the F4/80[hi] pool was an age related phenomena, the rate of which was not accelerated by nematode infection. This data is supported by the findings of *Bain et al., 2016* showing a gradual replenishment of the pleural F4/80[hi] population by bmMΦ with age (*Bain et al., 2016*). In that study, the level of bone marrow cell contribution to the pleural MΦ compartment is ~50% by 19 weeks of age. In our experiments, experimental mice were approximately 18 weeks of age at the time of infection. Thus by the time the mice were infected with *L. sigmodontis*, a substantial proportion of the F4/80[hi] population had already been replenished from the bone marrow. Because the ratio of recruited to resident is identical in naïve vs infected animals, throughout the infection time course, the far greater resMΦ numbers in C57BL/6 mice are not due to infection-driven recruitment of bone-marrow derived cells. Instead, higher F4/80[hi] numbers presumably result from in situ proliferation of the resMΦ population in infected mice regardless of origin.

All of F4/80[hi] MΦ were positive for residency markers GATA6 and CD102 (*Figure 2—figure supplement 1*). Comparison of GATA6 and CD102 expression by donor- and host-derived F4/80[hi] MΦ within naïve and infected C57BL/6 mice revealed that donor-derived cells were equally capable of expressing these residency markers as host-derived cells (*Figure 3D and E*, *Figure 3—figure supplement 2C–E*). Thus successful integration of bmMΦ cells into the resMΦ niche is exemplified by expression of GATA6 and CD102, a process that is not altered by infection. The situation was different for Tim4, also considered a marker of resident F4/80[hi] MΦ within the pleural space (*Davies et al., 2011*). Unlike GATA6 and CD102, which uniformly marked the F4/80[hi] population, only 46 ± 1.6% of the F4/80[hi] population was TIM4[+] in naïve animals aged 23–25 weeks (*Figure 3F*). In naïve controls at 25 weeks of age, an equal proportion of donor and host F4/80[hi] cells were negative for Tim-4. This data is consistent with the finding of *Bain et al., 2016* that even long-lived bone marrow derived cells do not universally take on Tim4

expression in the steady state (*Bain et al., 2016*). The striking result here was that *L. sigmodontis* infection induced TIM4 expression on the F4/80[hi] population, with 76 ± 4% staining TIM4 positive at day 35 and 50 pi (*Figure 3F*), and by day 50 Tim4 expression more closely resembled that of GATA6 and CD102 with the majority of both donor and host F4/80[hi] cells expressing Tim4 (*Figure 3F&G*). The data suggest that Tim4, both in the steady state but particularly during infection is not a reliable marker of MΦ origin.

## Macrophages from BALB/c mice exhibit enhanced PD-L2 expression, which associates with reduced worm killing

*Bain et al., 2016* have recently highlighted that in naïve mice RELMα is expressed by bone marrow precursors (F4/80[lo]MHCII[+]CSF1R[+]) to the F4/80[hi] resMΦ pool and that RELMα transiently marks cells of monocytic origin. Consistent with this finding, we observed ~60 and~80% of the F4/80[lo] MΦ were RELMα positive in naïve C57BL/6 and BALB/c mice respectively, while ~10% and ~50% were RELMα positive in the respective F4/80[hi] populations (*Figure 4A*). This data along with dimensionality reduction analysis of multi-parametric flow cytometry (*Figure 2—figure supplement 2*) illustrated fundamental differences in the dynamics of incoming cells between the strains. Downregulation of RELMα may be part of the process by which F4/80[lo]MHCII[+]CSF1R[+]MΦ convert to F4/80[hi] resMΦ in naïve animals. If so, the higher RELMα positivity within the F4/80[hi] population of BALB/c mice may reflect a relatively poor ability of BALB/c mice to promote bmMΦ integration into the resMΦ niche. However, upon infection of both strains almost all MΦ regardless of phenotype expressed RELMα (*Figure 4A*; *Figure 2—figure supplement 2*), a reflection of the ability of IL-4Rα signaling to induce RELMα expression independent of strain or origin (*Jenkins et al., 2013*). RELMα was therefore not a useful marker of monocytic origin in the context of type two immunity.

Like RELMα, the immunosuppressive molecule programmed cell death ligand 2 (PD-L2) is induced by IL-4 but in contrast to RELMα, is preferentially expressed by F4/80[lo] M(IL-4) (*Gundra et al., 2014*). In addition, PD-L2 is permanently downregulated by F4/80[lo] M(IL-4) upon integration into the resident pool even in the face of subsequent IL-4Rα signaling (*Gundra et al., 2017*). We therefore assessed the expression of PD-L2 at day 50 pi in both resistant C57BL/6 and susceptible BALB/c mice. PD-L2 expression was upregulated by all MΦ sub-populations of infected BALB/c mice compared to naïve controls (*Figure 4B*) whereas infected C57BL/6 mice exhibited significantly lower frequencies of PD-L2[+] MΦs relative to BALB/c mice (*Figure 4B*). Indeed, there was negligible detection of PD-L2 expression by the F4/80[hi] population of C57BL/6 mice at day 50 pi (*Figure 4B*), despite being ~80% bone marrow derived (*Figure 3B*). This provides additional evidence that successful integration into the resident niche results in an inability to upregulate PD-L2 in response to helminth infection, consistent with recent findings (*Gundra et al., 2017*). Thus, PD-L2[+] cells within the F4/80[hi] population of infected BALB/c mice supports a model of inefficient integration of bmMΦ into the resMΦ niche in this strain.

PD-L2 has specific relevance to *L. sigmodontis*, where its ligand PD-1 is highly expressed on T$_H$2 cells during chronic infection of susceptible BALB/c mice. In this setting, PD-1/PD-L2 interaction induces a state of T$_H$2 cell-intrinsic hypo-responsiveness that is characterised by significantly diminished production of T$_H$2 signature cytokines IL-4 and IL-5 (*van der Werf et al., 2013*). To assess if the enhanced recruitment and subsequent M(IL-4) activation of F4/80[lo] MΦ and monocytes was contributing to susceptibility through the induction of hypo-responsive T$_H$2 cells, the strength of the immune response was analysed at day 50 pi in both resistant C57BL/6 and susceptible BALB/c mice. Expression of the T$_H$2 master transcription factor GATA3 was significantly greater in CD4+ T cells isolated from the pleural cavity of C57BL/6 mice compared with BALB/c (*Figure 4C*) although no significant difference in expression of IL-4 or IL-5 by GATA3[+]CD4[+] cells was detected (*Figure 4C and D*). Interestingly however, there was a significantly greater proportion of IFN-γ producing CD4[+] cells from infected BALB/c mice compared to C57BL/6 mice (*Figure 4D*). Notably, there was a strong positive correlation between the percentage of PD-L2[+] MΦ and worm recovery rate (*Figure 4E*).

## F4/80[lo] MΦ and monocytes are detrimental to nematode killing

The data thus far suggested that in BALB/c mice recruitment of bmMΦ was detrimental to parasite killing. To test this hypothesis we used a monocyte depleting anti-CCR2 antibody to block monocyte recruitment in BALB/c mice prior to the peak of monocyte influx. Anti-CCR2 was administered daily

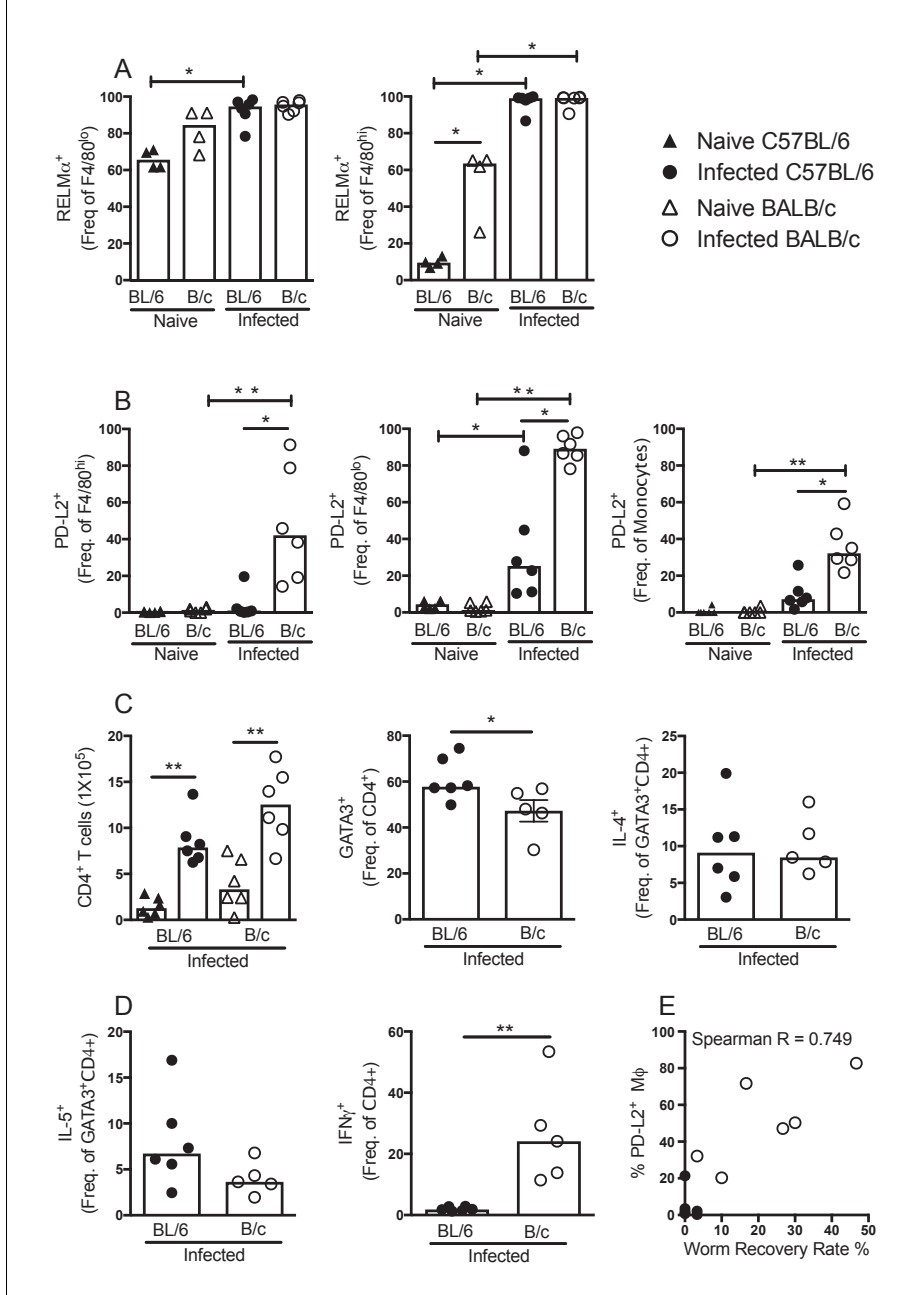

**Figure 4.** PD-L2+ MΦ are associated with susceptibility. Expression of (**A**) RELMα or (**B**) PD-L2 by F4/80[hi], F4/80[lo] and/or monocyte populations isolated from naïve and *L. sigmodontis* infected C57BL/6 and BALB/c mice at day 50 pi. (**C**) The number of CD4+ T cells and percentage of CD4+ T cells expressing GATA3, IL-4, (**D**) IL-5 and IFNγ within the pleural cavity (PC) of mice in (**A**). (**E**) Correlation between percentage of PD-L2 positive MΦ and worm recovery rate. Data in (**A,C,D**) are representative of three independent experiments with 6 mice/group, data in (**B**) is representative of two independent experiments. *p<0.05, **p<0.01 (Non-parametric Kruskal-Wallis test preformed followed by Mann–Whitney for pairwise comparison). Bars represent the median.
DOI: https://doi.org/10.7554/eLife.30947.010

The following figure supplement is available for figure 4:

**Figure supplement 1.** Monocyte depletion does not alter the accumulation of PD-L2+ MΦ.
DOI: https://doi.org/10.7554/eLife.30947.011

from day 31–34-post infection and PLEC were examined at day 35 pi (*Figure 5A*). Antibody treatment in this time frame successfully prevented the influx of monocytes into the pleural cavity, resulting in the F4/80$^{hi}$ population representing ~80% of the total MΦ compartment (*Figure 5B*). Inhibiting monocyte infiltration resulted in significantly increased worm killing in the normally susceptible BALB/c mice (*Figure 5C*). In the anti-CCR2 treated group 29% of the mice had no parasites compared to 5% in the rat IgG treated group, a striking effect given the BALB/c mice typically have not cleared the infection until ~day 90 pi. Anti-CCR2 treatment did not affect the number of F4/80$^{hi}$ MΦ in the cavity, but significantly reduced F4/80$^{lo}$ MΦ and depleted monocytes (*Figure 5D*). The difference in parasite killing could not readily be attributed to PD-L2, however, as there was no significant difference in the percentage or number of PD-L2$^{+}$ MΦ between anti-CCR2 treated and Rat IgG treated controls at this time point (*Figure 5E* and *Figure 4—figure supplement 1*). While no difference was detected in the number of pleural cavity CD4$^{+}$GATA3$^{+}$ T$_{H}$2 cells between control and monocyte depleted mice (*Figure 5F*), the proportion of CD4$^{+}$GATA3$^{+}$T$_{H}$2 cells producing IL-4 in monocyte depleted mice was significantly enhanced (*Figure 5F*). The frequency of IL-5$^{+}$ and IFN-γ$^{+}$ cells were not significantly altered (*Figure 5F*).

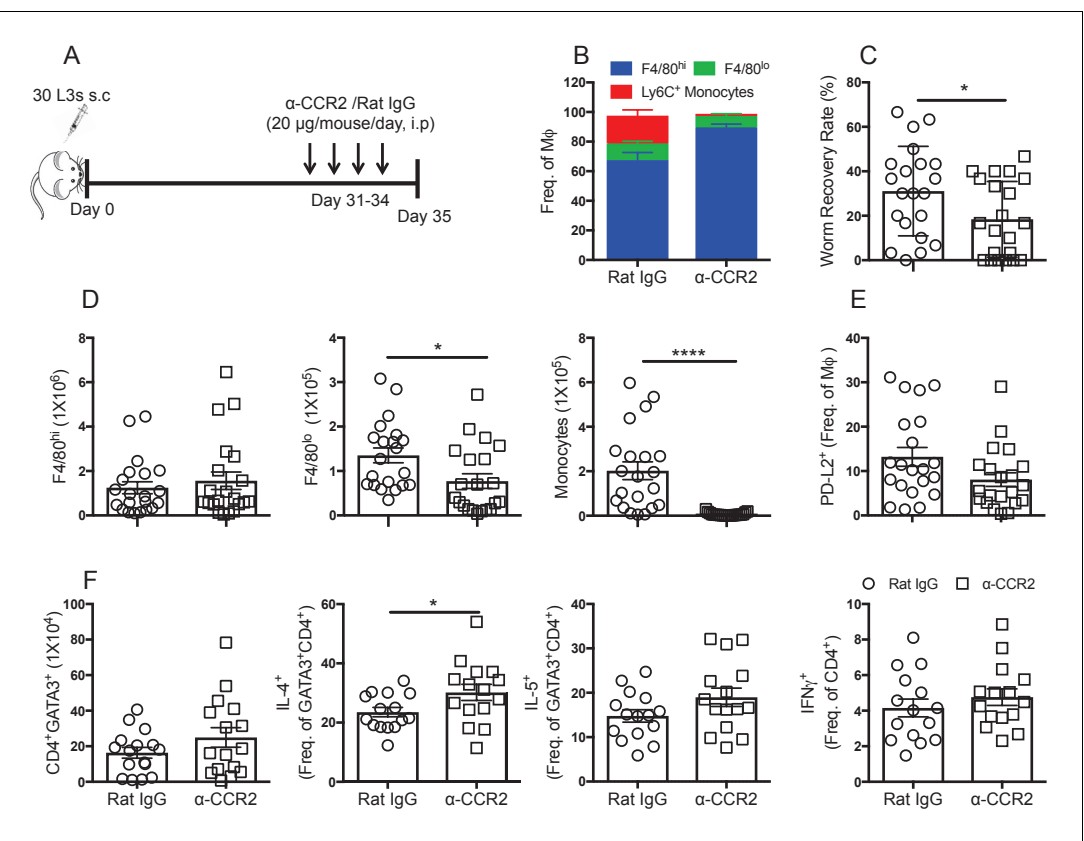

**Figure 5.** Recently recruited MΦ are detrimental to worm killing. (**A**) Experimental scheme. (**B**) Contribution of F4/80$^{hi}$, F4/80$^{lo}$ and monocytes to the MΦ compartment of *L. sigmodontis* infected BALB/c mice after treatment with either Rat IgG (circles) or anti-CCR2 (squares). (**C**) Worm recovery rate at day 35 pi following 4 days of either control rat IgG or α-CCR2 administration i.p. in susceptible BALB/c mice. (**D**) Pleural F4/80$^{hi}$, F4/80$^{lo}$, monocyte numbers. (**E**) Percentage of MΦ expressing PD-L2. (**F**) Total number of pleural cavity GATA3$^{+}$CD4$^{+}$ T cells, percentage GATA3$^{+}$CD4$^{+}$ expressing IL-4 or IL-5 and IFNγ expression by CD4$^{+}$ cells. (**C and D**) *p≤0.05, ****p≤0.0001, as determined by an ANOVA using combined data from three experiments with 5, 10 and 6 mice/ group (**F**) *p≤0.05 determined by an ANOVA using data combined from two experiments with 10 and 6 mice/ group (p<0.05, Tukey's HSD).
DOI: https://doi.org/10.7554/eLife.30947.012

## Discussion

Through comparison of resistant and susceptible strains during filarial infection we demonstrate that dichotomous functions attributed to M(IL-4) may be explained by whether a MΦ possesses a recently recruited bmMΦ or resMΦ phenotype upon IL-4Rα stimulation. We also highlight striking differences in the dynamics of serous cavity MΦs between two commonly used laboratory mouse strains. Resistance against *L. sigmodontis* in C57BL/6 mice was associated with the predominance of an M(IL-4) F4/80$^{hi}$GATA6$^+$CD102$^+$ resMΦ population that accumulated through proliferation of the existing population. The resMΦ phenotype within the serous cavities is in part determined by the retinoic acid-dependent master transcription factor GATA6 and additional retinoic acid independent genes such as CD102 (*Okabe and Medzhitov, 2014*; *Rosas et al., 2014*; *Bain et al., 2016*). Here we show that bmMΦ successfully integrated into the resMΦ niche, as defined by GATA6 and CD102 expression in both naïve and infected animals. This data re-enforces the finding that resMΦ of the serous cavity are replenished from the bone marrow in an age-dependent manner (*Bain et al., 2016*). We further show that despite a dramatic increase in MΦ number within the cavity as well as infection-dependent changes in phenotype, helminth infection does not alter the homeostatic rate of age-dependent replenishment. The data suggests that regardless of the size of the macrophage pool inherent differences in proliferation, survival and death between long-term and recent entrants into the resident pool can maintain the ratio of bone-marrow to embryonically-derived cells in the pleural cavity (*Supplementary file 1*).

Resistance was also associated with a greater degree of pleural B cell accumulation. B cells and their antibody products play a central role in anti-nematode immunity (*Esser-von Bieren et al., 2013*; *Rajan et al., 2005*; *Carter et al., 2007*) and we recently reported that during primary *L. sigmodontis* infection, pleural cavity B cells proliferate and produce antigen-specific IgM (*Jackson-Jones et al., 2016*). Collaboration between MΦ and antibodies to trap and kill invading larvae has been demonstrated during secondary *H. polygurus* infection (*Esser-von Bieren et al., 2013*). Experiments in a related model of filarial nematode infection in the peritoneal cavity demonstrate that resident MΦ actively contribute to larval death (unpublished data). Consequently, we hypothesise that local IgM and MΦ together function in granuloma formation, eventually leading to death of the *L. sigmodontis* worm in resistant C57BL/6 mice.

Susceptibility in the BALB/c strain was marked by significantly less F4/80$^{hi}$ macrophage proliferation. Additionally, the F4/80$^{hi}$GATA6$^+$CD102$^+$ resMΦ population within both naïve and infected BALB/c mice diminished with age while the percentage of bmMΦ contributing to the MΦ compartment increased. Such dynamics are suggestive of an inability of influxing bmMΦ to integrate into the resMΦ niche of BALB/c mice and could reflect a deficit in local retinoic acid. Indeed a recent study has highlighted the inability bmMΦ to integrate into the resMΦ niche of C57BL/6 mice on a Vitamin A deficient diet (*Gundra et al., 2017*). Alternatively bmMΦ in BALB/c mice may be transcriptionally silenced at the GATA6 locus as has been noted for thioglycollate elicited MΦ (*Okabe and Medzhitov, 2014*).

Infection in the BALB/c strain was also marked by an influx of Ly6C$^+$ monocytes by day 35 pi. resulting in a large proportion of the myeloid compartment possessing a recently recruited F4/80$^{lo}$MHCII$^{hi}$ phenotype by day 50 pi. Because these cells preferentially express PD-L2 (*Gundra et al., 2014*), there was also a greater degree of PD-L2 expression upon M(IL-4) activation. Monocyte depletion prior to day 35 pi revealed an immunosuppressive role for monocytes and F4/80$^{lo}$ M(IL-4) during *L. sigmodontis* infection, with an enhanced T$_H$2 profile.

While our data demonstrate a role for the incoming monocytes in the susceptibility of BALB/c mice, we are not asserting that differences in monocyte recruitment are the sole reason for differences in strain susceptibility. We hypothesise that susceptibility in the BALB/c strain arises from a combination of factors, beginning with a failure to generate resMΦ and B cell numbers equivalent to that seen in the resistant strain. Secondarily, we hypothesize that this deficit in resMΦ and B cells is confounded by an influx of bone-marrow derived macrophages which assume an immunosuppressive PD-L2+ phenotype which fail to integrate into the resident niche, thereby affecting the ability of MΦ to function in an anti-helminthic manner. A role for PD-L2 expressing cells in mediating susceptibility in BALB/c mice during *L. sigmodontis* infection has already been demonstrated (*van der Werf et al., 2013*) and the immunosuppressive monocytes we observe likely contribute to the T regulatory response that is important for susceptibility to *L. sigmodontis* (*Taylor et al., 2005*; *Taylor et al.,*

2007; Taylor et al., 2009). Notably, the effects of Treg depletion while significant are similarly modest to the effects seen here with monocyte depletion further suggesting that Treg expansion and immunosuppressive monocytes are not alone responsible for susceptibility.

It is notable that a similar influx of monocytes that mature into PD-L2$^+$ bmMΦ is seen in the liver of *S. mansoni* infected mice (*Pearce and MacDonald, 2002*; *Nascimento et al., 2014*; *Gundra et al., 2014*). While depletion of this immunosuppressive bmMΦ population during *S. mansoni* infection results in an enhanced T$_H$2 immune response it is also marked by reduced granuloma formation and severely exacerbated disease (*Nascimento et al., 2014*). A direct host-protective role of these cells is illustrated in a study whereby PD-L2$^+$ bmMΦ isolated from *T. crassiceps* infected mice reduce the disease burden during experimental autoimmune encephalitis (*Terrazas et al., 2017a*). In both studies the immunosuppressive impact of PD-L2$^+$ bmMΦ were observed late in infection. Although PD-L2 inhibits protective immunity to *L. sigmodontis* (*van der Werf et al., 2013*), we were unable to demonstrate that the immunosuppressive capacity of bone-marrow derived M(IL-4) at this time point was due to PD-L2, and suspect that day 35 is too early to see a PD-L2 dependent effect.

Whether eventual parasite death within the BALB/c strain results because of a decline in immunosuppressive bmM(IL-4) or successful integration of these cells into the resMΦ niche remains to be explored. Indeed, as *S. mansoni* infection progresses, PD-L2$^+$ bmMΦ integrate into the resident niche enabling disease resolution. In the absence of vitamin A, integration into the resident niche is inhibited and enhanced mortality pursues (*Gundra et al., 2017*). Together with our data these studies suggest that monocyte-derived macrophages are critical regulators of host immunity, which can tip the fine balance between infection control and host damage. As such we hypothesise that bmMΦ upon infiltration into an inflammatory milieu containing IL-4 take on an immunosuppressive phenotype, to prevent self-damage associated with more classical inflammatory pathways. Of note, similar differences in monocyte recruitment might contribute to susceptibility and resistance between BALB/c and C57BL/6 mice to *Leishmania donovani*. A recent publication by (*Terrazas et al., 2017b*) demonstrated that CCR2 dependent monocytes contributed to host susceptibility in BALB/c mice. In contrast, *Sato et al., 1999* found no role for CCR2 in mediating resistance in C57BL/6 mice, suggesting that the difference we observed here may be a fundamental difference between these strains that goes beyond helminth infection.

## Materials and methods

### Mice and experimental grouping

Female BALB/c mice, congenic CD45.1$^{+/+}$ / CD45.2$^{+/+}$ C57BL/6 mice, and C*cr2*-/-mice were bred in house and maintained in specific pathogen free (SPF) facility at the University of Edinburgh. *Ccr2*-/-mice were originally sourced from Jackson Laboratories (C57BL/6 *Ccr2*$^{tm1lfc}$). Experimental animals were 6–8 weeks of age at the beginning of the experiment, unless otherwise stated. Age matched female mice were randomly allocated into naïve and infected groups on day 0 of an experiment. Naïve and infected mice were housed in separate individually ventilated cages. Sample size was calculated on the basis of the number of animals needed for detection of a change in macrophage proliferation of 50% at a P value of < 0.05, using our experience with the *L. sigmodontis* model (*Jenkins et al., 2011*; *Jenkins et al., 2013*). Due to the inherent biological variation in the parasite lifecycle, the number of available infective nematode larvae determined the number of time points that could be examined in a single experiment. Experiments were in accordance with the United Kingdom Animals (Scientific Procedures) Act of 1986 and approved by the University of Edinburgh Animal Welfare and Ethical Review Body.

### Generation of partial bone marrow chimeric mice

CD45.1$^{+/+}$ C57BL/6 females were anaesthetized and their lower limbs exposed to 9.5 Gy γ-irradiation. The upper body, including the pleural cavity was protected by a two inch lead shield. Partially irradiated animals were administered 4.55 × 10$^6$ CD45.2$^{+/+}$ bone marrow cells intravenously. Recipient mice were allowed to recover for 8 weeks prior to *L. sigmodontis* infection.

### Litomosoides sigmodontis infection

*L. sigmodontis* infective stage 3 larvae (L3) were isolated from the mite vector *Ornithonyssus bacoti*. Thirty L3s suspended in 200 μl RPMI (5% horse serum) were then injected subcutaneously into the scruff using a 23G needle.

### Anti-CCR2 MAb administration

For monocyte depletion 20 μg of MC-21 (*Mack et al., 2001*; RRID:AB_2314128) was administered by intraperitoneal injection from day 31–34 pi prior to termination of the experiment at day 35 pi. Control animals were similarly injected with Rat IgG (BioXcell, UK, Clone: LTF2)

### Cell isolation

Pleural exudate cells (PLEC) were obtained through washing the pleural cavity with 10 ml RPMI supplemented with penicillin-Streptomycin (1%) and L-Glutamine (1%). Samples were kept on ice and contaminating erythrocytes were lysed prior to cell counting with a Nexcelom cell counter.

### Flow cytometry

Pleural cells ($5 \times 10^5$/100 μl or $1 \times 10^6$/200 μl) were washed twice in PBS, stained with LIVE/DEAD (Invitrogen, Carlsbad, CA) and blocked with 0.025 μg anti-CD16/32 (2.4G2: Biolegend, San Diego, CA) and 1:100 heat-inactivated mouse serum (Invitrogen) prior to surface staining; CD19 (6D5), Ly6G (1A8), SigLecF (E50-2440), TCRβ (H57-597), MHC class II (M5/115.15.2), F4/80 (BM8), Ly6C (HK1.4), CD115 (AFS98), CD11b (M1/70), CD11c (N418), CD102 (3C4(MIC2/4)), PD-L2 (TY25), CD4 (GK1.5). Samples were washed, permeabilized overnight (FoxP3/Transcription Factor Staining Buffer Set, eBioScience, San Diego, CA) and stained for intracellular marker GATA6 (D61E4), Ki67 (REA183), GATA3 (REA174), IL-4 (11B11), IL-5 (TRFK5), IFNγ (XMG1.2), YM1 (DY2446, R&D Systems, Wiesbaden, Germany) or RELMα (PeproTech, Rocky Hill, NJ). Where necessary samples were stained with streptavidin and anti-rabbit conjugated fluorochromes. Cells were acquired using the FACS LSR Fortessa with FACSDiva software. FlowJo version nine software was used for data analysis. For dimensionality reduction analysis of monocyte/macrophage populations, pleural cavity cells from day 35-infected and naïve mice were stained as above. Using Flowjo, lineage negative, CD11b$^+$cells were concatenated and exported from 5 mice per group and down-sampled to 10,000 cells. tSNE and PCA dimensionality reduction using 13 parameters, (Ly6C, YM1, CD11b, CD115, PDL-2, GATA6, TIM4, RELMA, MHC-II, CD11c, F4/80 forward-scatter area and side scatter area) was performed using the Bioconductor R package, Cytofkit and FCS files were exported for analysis in FlowJo.

### Statistical analysis

Statistical significance on data from naïve and infected C57BL/6 and BALB/c mice, was carried out using a two way-ANOVA. When data was combined from multiple experiments, experimental effects were controlled for in the analysis. Where a dataset failed to meet the requirements for a parametric test, comparison was performed with a non-parametric unpaired Mann-Whitney-Wilcoxon. GraphPad Prism v6.0 and JMP version 12 were used for the statistical tests.

### Exclusion criteria

One animal was excluded from the cytokine analysis graphs of *Figure 4*. The majority of cells from this sample were dead post PMA + Ionomycin restimulation, as evidenced on the flow cytometer by ~531 CD4$^+$GATA3$^+$ events compared with ~5000 events in other samples. A threshold was set at 1000 events because with fewer events the percentages were not representative.

## Acknowledgements

The authors would like to extend thanks to Alison Fulton and Sheelagh Duncan for excellent technical assistance, Dr. Martin Waterfall for flow cytometry support and the University of Edinburgh veterinarian and support staff for excellent animal husbandry. SC was the recipient of a principal career development PhD scholarship provided by the University of Edinburgh. This work was supported by MRC-UK grants to JEA [MR/K01207X/1], MDT [MR/K020196/1] and SJ [MR/L008076/1].

# Additional information

### Funding

| Funder | Grant reference number | Author |
|---|---|---|
| Medical Research Council | MR/K01207X/1 | Judith Allen |
| Medical Research Council | MR/K020196/1 | Matthew Taylor |
| University Of Edinburgh | Principal Career Development PhD Scholarship | Sharon Campbell |
| Medical Research Council | MR/L008076/1 | Stephen Jenkins |

The funders had no role in study design, data collection and interpretation, or the decision to submit the work for publication.

### Author contributions

Sharon M Campbell, Conceptualization, Data curation, Formal analysis, Investigation, Visualization, Methodology, Writing—original draft; Johanna A Knipper, Conceptualization, Formal analysis, Investigation, Visualization, Writing—review and editing; Dominik Ruckerl, Stephen J Jenkins, Conceptualization, Supervision, Writing—review and editing; Conor M Finlay, Formal analysis, Investigation, Writing—review and editing; Nicola Logan, Carlos M Minutti, Investigation; Matthias Mack, Resources; Matthew D Taylor, Conceptualization, Funding acquisition, Writing—review and editing; Judith E Allen, Conceptualization, Supervision, Funding acquisition, Writing—original draft, Project administration

### Author ORCIDs

Sharon M Campbell (iD) https://orcid.org/0000-0002-5558-0749
Johanna A Knipper (iD) https://orcid.org/0000-0002-9038-0566
Dominik Ruckerl (iD) http://orcid.org/0000-0002-0206-1451
Carlos M Minutti (iD) https://orcid.org/0000-0002-9663-1928
Judith E Allen (iD) http://orcid.org/0000-0002-3829-066X

### Ethics

Animal experimentation: All animal experiments were performed in accordance with the UK Animals (Scientific Procedures) Act of 1986 under Project Licenses (60/4104 & 70/8548) granted by the UK Home Office and approved by the University of Edinburgh and University of Manchester Animal Welfare and Ethical Review Body. Partial bone marrow chimeras were performed under anaesthesia (Ketamine/Medetomidine; 1/1; v/v) and every effort was made to minimise suffering.

### Decision letter and Author response

Decision letter https://doi.org/10.7554/eLife.30947.017
Author response https://doi.org/10.7554/eLife.30947.018

# Additional files

### Supplementary files

• Supplementary file 1. Schematic of propsed MΦ dynamics. (A) Post birth, prenatally derived F4/80$^{hi}$ resMΦ constitute 80% of the MΦ compartment, by 18–23 weeks of age 50% of the resMΦ compartment has been replenished by bmMΦ which assume residency markers GATA6 and CD102. The degree of bmMΦ contributing to the resMΦ population continues to increase with age, reflective of enhanced proliferative survival of the more recent donor derived resMΦ and eventual death of host derived F4/80$^{hi}$ resMΦ. (B) Upon infection, IL-4 drives proliferation of F4/80$^{hi}$ resMΦ, 50% of which has been derived from bmMΦ by 23 weeks of age, causing expansion of the resMΦ population to 27-fold greater than that of naïve controls. By day 50 pi, the degree of bmMΦ contributing

to the resMΦ population has increased further, likely reflecting enhanced proliferative survival of the more recent donor derived resMΦ and eventual death of host derived F4/80$^{hi}$ resMΦ
DOI: https://doi.org/10.7554/eLife.30947.013

• Transparent reporting form
DOI: https://doi.org/10.7554/eLife.30947.014

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
