## [Decision Letter]

Thank you for submitting your article "Myeloid cell recruitment versus local proliferation differentiates susceptibility from resistance to filarial infection" for consideration by *eLife*. Your article has been reviewed by 3 peer reviewers, and the evaluation has been overseen by a Reviewing Editor and Michel Nussenzweig as the Senior Editor. The following individual involved in review of your submission has agreed to reveal his identity: Steffen Jung (Reviewer #1).

The reviewers have discussed the reviews with one another and the Reviewing Editor has drafted this decision to help you prepare a revised submission.

Summary:

In this work, killing of filarial nematodes the pleural space in dependence of IL-4 activated macrophages (MΦ) was investigated in two mouse strains with graded susceptibility. The investigators show that MΦ proliferation in situ was a dominant cause of MΦ expansion in infected resistant B6 mice. In contrast, BALB/c mice showed a reduced population of resident MΦ, but an increased influx of immunosuppressive PD-L2^+^ monocytes, which inhibited parasite killing. Overall, this is very interesting work providing in vivo evidence that the myeloid cell composition in the pleural space is essential for parasite killing, and that incoming monocytes impair resistance. It is particularly interesting that BALB/c mice fail to integrate bone-marrow derived MΦ into the residency pool.

The presented experiments are convincing and scientifically sound.

The paper does not provide a clear-cut mechanism by which the differential representation of the two macrophage populations (F4/80 high vs. low) mediates resistance and susceptibility to infection, respectively. The study remains correlative and does not offer any data on effector functions of the two macrophage populations. Also, the work does not exclude the possibility that the differential outcome of infection is related to indirect/secondary effects of the macrophage populations. The putative direct impact of the macrophage populations on the protective Th2 response remained rather weak and is unlikely to fully explain the differential clinical phenotype of BALB/c and C57BL/6 mice. These alternative options should be discussed.

Essential revisions:

1) The phenotypic differences between B6 and BALB/c mice are complex due to the different development of the parasites in these mice. As outlined in the Introduction, in susceptible BALB/c mice, microfilariae are produced and circulate in the bloodstream of B/c mice starting at 55 days post infection. In contrast to BALB/c mice, microfilariae are not produced in B6 mice. The differences between the mouse strains in the pleural macrophage dynamics are striking. Yet, I am not fully convinced that the late monocyte influx in BALB/c mice is the major difference between the mouse strains, since the effect of the CCR2 mAb is rather modest.

Furthermore, the paper would be stronger, if effects were always shown site by site between the strains.

2) The dynamics in total pleural cell numbers in infected mice can only partially be explained by changes in the myeloid cells and B cells. As an example at day 28, macrophages, monocytes and B-cells sum up to around 7 x 106 out of 20 x 106 cells. What are the other cells?

3) I am missing monocyte subset definition. Are all monocytes Ly6Chi?

4) Please show neutrophil influx during infection: less than 1% of cells correspond to the monocyte influx early in infection.

5) The expression of CD102 and GATA6 is hard to grasp from the provided data. Please provide original dot blots of day 50 mice with and without infection (supplementary data).

6) The data presented in Figure 3 are incomplete, i.e. not conclusive in the current form. The kinetics of Ki67 staining does not follow that of the cellular expansion. Please provide additional data, in particular BrDU staining.

7) The half-life of pleural MΦ in the bone marrow chimeras is not plausible. The half-life is around 13 weeks at day 35 (8 weeks resting plus 5 weeks experiment), but 2 weeks between day 35 and day 50. How is that?

8) Figure 3: Please show original dot blots in the supplement.

9) Figure 4.: Whereas the data on PD-L2^+^ monocytes are intriguing, the interpretation of the T-cell data is not acceptable. Please remove the statement on a “slight trend", there is no difference in T-cells, IL5^+^CD4^+^ cells and GATA3^+^ CD4 cells between the strains.

10) The numbers of mice from the same condition differ between the panels, e.g. Figure 4 or 6 infected BALB/c mice? Why is this? Please explain in detail.

11) Figure 5. Please eliminate the statement on a trend (see above).

12) Statistics: Please prove normal distribution of values or use a non-parametric test. In this regard I am particularly doubtful about Figure 5! In the same panel: If the authors wish to show that more mice with intervention than with the control cleared the infection – as elaborated in the text – a completely different analysis is required. Otherwise, please remove the statement and adjust the statistics.

13) Figure 1: The authors state that the presented data are pooled from two separate time course experiments (day 11 + 28 and day 35 + 50). However, the data are presented as continuous line graphs, which should only be used if all data points are from one time course experiment. Please change into bar graph and explicitly label column 1 + 2 as "experiment 1" and column 3 + 4 as "experiment 2".

14) It remains unclear from the text and Figure 3—figure supplement 2, why the authors chose a partial irradiation of the mice, in which not only the chest and abdomen, but also the upper extremities were protected against irradiation. Essentially, this means that the recipient mouse has quite significant residual bone marrow activity, as hematopoiesis is not restricted to long bones of the lower extremities and also take places in the upper extremities, sternum, pelvis etc. In order to calculate the contribution of bone derived-macrophages vs. resident tissue macrophages to the cellular exudates in the pleural cavity, it would have been sufficient to shield the chest alone in order to minimize bone marrow activity derived from the recipient; the donor (BM-derived) and recipient (tissue-resident) macrophages can be distinguished just by the congenic CD45 alleles. As it stands, the authors need to better specify and explain the contribution of CD45.2^+^ (derived from donor bone marrow) and CD45.1^+^ macrophages (derived from residual recipient bone marrow) to the macrophage pool in the pleural space. Most importantly, the authors wanted to establish, whether the increased F4/80high population at day 35 and 50 post infection was the result of local F4/80high proliferation or of a recruitment and conversion of monocytes into the F4/80high pool (subsection “Resistant C57BL/6 mice maintain the F4/80^hi^ resMΦ population”, first paragraph). There is no clear answer to this question in this subsection, and probably also not possible due to the fact that recipient bone-marrow-derived macrophages expressing high amounts of F4/80 cannot be distinguished from F4/80high macrophages derived from the resident tissue macrophage population.

15) Has it ever been tested whether the resident tissue macrophages in the pleural space are radiosusceptible or radioresistant?

---

## [Author Response]

Essential revisions:1) The phenotypic differences between B6 and BALB/c mice are complex due to the different development of the parasites in these mice. As outlined in the Introduction, in susceptible BALB/c mice, microfilariae are produced and circulate in the bloodstream of B/c mice starting at 55 days post infection. In contrast to BALB/c mice, microfilariae are not produced in B6 mice. The differences between the mouse strains in the pleural macrophage dynamics are striking. Yet, I am not fully convinced that the late monocyte influx in BALB/c mice is the major difference between the mouse strains, since the effect of the CCR2 mAb is rather modest.

We are in agreement with the reviewer. This is a highly complex chronic infection with a metazoan parasite. It is certain that multiple factors contribute to resistance or susceptibility and we do not believe a single cellular population can explain susceptibility. Indeed, previous papers demonstrating a significant role for Treg cells in susceptibility to *L. sigmodontis* similarly do not show dramatic changes – nonetheless they are significant and consistent effects [1-3]. We hypothesize that susceptibility in the BALB/c strain arises from a combination of factors, beginning with a failure to generate resMΦ and B cell numbers equivalent to that seen in the resistant strain. Secondarily, we hypothesize that this deficit in resMΦ and B cells is confounded by an influx of bone-marrow derived macrophages which assume an immunosuppressive PD-L2^+^ phenotype and fail to integrate into the resident niche, thereby affecting the ability of MΦ to function in an anti-helminthic manner. A clear role for PD-L2 expressing cells in mediating susceptibility in BALB/c mice during *L. sigmodontis* infection has already been demonstrated [4]. Critically, we are not asserting that differences in monocyte recruitment are the most important or sole reason for differences in strain susceptibility. We have worked to clarify the text in the Discussion to make this distinction clear and highlight the concept that multiple cell types must be contributing to resistance vs. susceptibility. Nonetheless, the manuscript does highlight real differences between resident and monocyte-derived cells in a relevant infection model. We further believe the great strength of the manuscript is in demonstrating that different genotypes exhibit very different myeloid recruitment/proliferation dynamics with an impact on infection outcome.

Furthermore, the paper would be stronger, if effects were always shown site by site between the strains.

We are not certain what is being requested here. The only site in the paper is the pleural cavity. If the referee meant side by side, we believe that we have done this. We have made it clearer in the text and figure legend that each figure is referring to the pleural cavity.

2) The dynamics in total pleural cell numbers in infected mice can only partially be explained by changes in the myeloid cells and B cells. As an example at day 28, macrophages, monocytes and B-cells sum up to around 7 x 106 out of 20 x 106 cells. What are the other cells?

This is a very relevant question and we apologise that we did not make this clearer. All of the populations contributing to total cell number are now represented in Figure 1—figure supplement 1. We had originally stated ‘Neutrophils were also significantly higher in C57BL/6 mice at d28 and d35 but represented less than 1% of the total population, while T cells and eosinophils did not differ significantly between strains (data not shown)’. However, we appreciate it was not clear that eosinophils, T cells and neutrophils constitute the remainder of the total cell number. Consequently, along with the new supplemental figure, we have edited the text to more clearly convey the contribution of different cell types to the total pleural cell numbers.

3) I am missing monocyte subset definition. Are all monocytes Ly6Chi?

We apologise for the confusion. Due to imprecision of our wording in the sentence below, we may have misled the referee into thinking that we had subdivided the monocyte population. We stated:

‘To further characterise the MΦ dynamics, we profiled the percentage contribution of F4/80^hi^, F4/80^lo^ and monocyte sub-populations to the MΦ compartment as a whole within each strain of naïve and infected animals’

The use of ‘sub-populations’ in this setting referred to total MΦ population; i.e. F4/80^hi^, F4/80^lo^ and monocytes. We have altered the text as follows:

‘To further characterise the MΦ dynamics within the pleural cavity, we profiled the percentage contribution of F4/80^hi^, F4/80^lo^ and monocyte populations to the MΦ compartment as a whole within each strain of naïve and infected animals.’

The following sentence was also added to more accurately convey the monocyte classification: ‘Thus the monocyte gate contained Lin-(CD19,Ly6G,SiglecF,TCRβ), CSF-1R^+^,F4/80lo,Ly6C^+^ cells.’

To further clarify the cell populations including Ly6C^hi^ expression, we are including multi-parameter flow cytometric data from a recent repeat of *L. sigmodontis* infection at Day 35. The Allen lab has moved from Edinburgh to Manchester and despite the change in animal unit and experimenter, the data has been highly repeatable. Indeed, the consistency of the findings is remarkable. We include the data as supplemental because the experiment was only performed once. Dimensionality reduction analysis of the flow cytometry data further illustrates the dramatic differences between the two strains, both in the steady state and at each infection time point (Figure 2—figure supplement 2).

4) Please show neutrophil influx during infection: less than 1% of cells correspond to the monocyte influx early in infection.

The neutrophil data can now be found in Figure 1—figure supplement 1.

5) The expression of CD102 and GATA6 is hard to grasp from the provided data. Please provide original dot blots of day 50 mice with and without infection (supplementary data).

The original dot plots of CD102 and GATA6 expression by the total MΦ population both in the presence and absence of infection were included in Figure 2—figure supplement 1. However, we recognize now that the labeling and layout of this figure was ambiguous. Figure 2—figure supplement 1 has been updated to more clearly illustrate the expression of GATA6 and CD102 by the total macrophage population and the F4/80^hi^ population in both strains in naïve animals and at day 11 and day 50 pi.

6) The data presented in Figure 3 are incomplete, i.e. not conclusive in the current form. The kinetics of Ki67 staining does not follow that of the cellular expansion. Please provide additional data, in particular BrDU staining.

We presume the referee is suggesting that we are missing proliferation by using ki67^hi+^ staining rather than BrDU. We do not believe this is the case. We have extensively verified that gating on Ki67^hi^ cells using BD clone BV56 (and only this clone), specifically marks cells actively in cell cycle, overlapping with BrDU staining following a 3 hour pulse. This was shown in our original J. Exp. Med and Science papers describing proliferation in this nematode model [5,6] and subsequently in our more recent Science paper where it can be clearly seen that all BrdU^+^ cells are contained within the Ki67hi population. Please see Suppl. Figure 3: www.sciencemag.org/content/356/6342/1076/suppl/DC1. The validity of using Ki67hi has been further confirmed independently by Phil Taylor [7] using pHH3 staining, which provides a definitive marker of proliferative events.

Of course, this does not resolve the concern of the referee about the apparent disconnect between the proliferation we observe and cellular expansion, as the increased number of F4/80^hi^ cells at day 28 and 35 could not be explained by Ki67^hi^ staining at these time points. Based on our experience with this model in which we have observed small proliferative resurgences, we assumed that proliferation occurred outside the time points examined. However, a detailed time course to identify transient proliferative peaks between day 11 and 35 post infection is not practical in terms of parasite availability or cost. It was exactly for this reason that we generated shielded bone marrow chimeras.

The use of the bone marrow chimeras allowed us to establish definitively whether the increase in number was due to local expansion or recruitment of bone marrow-derived cells. This is a well-established technique for this purpose [5,8-10].

We have now included data from CCR2 deficient mice(Figure 3—figure supplement 1), which was a major reason for our decision to proceed with chimeras. The increased F4/80^hi^ cell numbers within the pleural cavity of C57BL/6 mice at day 28 post *L. sigmodontis* infection, was unchanged in the CCR2 deficient animals supporting our hypothesis of local expansion.

An additional possibility, which may partially contribute to increased cell numbers is that in the context of infection fewer macrophages may be dying between d11 and later time points. In other words, even if proliferation is not higher than steady state levels at many time points, if IL-4 provides a survival signal and fewer cells die in that time frame, cell numbers would increase.

We have made an effort to clarify these points in the text and data from CCR2-/- C57BL/6 mice has been included in the form of Figure 3—figure supplement 1.

7) The half-life of pleural MΦ in the bone marrow chimeras is not plausible. The half-life is around 13 weeks at day 35 (8 weeks resting plus 5 weeks experiment), but 2 weeks between day 35 and day 50. How is that?

We too were surprised by the increased in BMD macrophage contribution between 23-25 weeks of age / 35-50 days pi.

We don’t believe the difference is due to some artifact of the system because there was no significant difference between the two time points in the blood chimerism on which the data is calculated.

The data instead suggests that during aging there is a point at which the contribution of monocyte derived cells increases sharply rather than being a steady accumulation over time. Bain et al. (2016) showed that replacement of pleural macrophages in naive B/6 mice is 25% at 12 weeks and 85% at 36 weeks, while we observe 45% at 23 weeks and 75% at 25 weeks. We hypothesise that there is a sigmoidal curve in which in the early time points there is a slow replacement of the resident cells with bone marrow derived cells and that this process increases sharply for a period in adulthood (e.g. 23-25 weeks) before leveling off near maximum.

Consistent with this finding is unpublished data from Stephen Jenkin’s lab in which bone marrow from young mice were transferred into young (8w red) and old (37 wk blue) mice. There was a striking increase in the replacement of pleural cavity F480hi cells in the older mice vs. the younger mice.

It is important to emphasise that regardless of the rate of resident macrophage turnover, our interpretation of the results still stand. The equal level of chimerism between naïve and infected animals at day 35 and day 50 supports the conclusion that increased cell number result from local expansion of F4/80^hi^ cells and not recruitment of bone marrow derived cells.

8) Figure 3: Please show original dot blots in the supplement.

These are now provided in the supplemental data (Figure 3—figure supplement 2).

9) Figure 4.: Whereas the data on PD-L2^+^ monocytes are intriguing, the interpretation of the T-cell data is not acceptable. Please remove the statement on a “slight trend", there is no difference in T-cells, IL5^+^CD4^+^ cells and GATA3^+^ CD4 cells between the strains.

Any reference to trends has been removed. However, our re-analysis of statistics (as requested below) revealed that the there is a statistical difference in the proportion of GATA3^+^ CD4^+^ cells between the strains. We have changed the text but have been careful not to overstate the conclusions.

10) The numbers of mice from the same condition differ between the panels, e.g. Figure 4 or 6 infected BALB/c mice? Why is this? Please explain in detail.

We thank the reviewer for drawing attention to this discrepancy. The count of total T cell number was carried out on samples prior to stimulation with PMA and Ionomycin. Following PMA and Ionomycin stimulation, one sample consisted primarily of dead cells, as seen on the flow cytometer and evidenced in a very low event count in the CD4^+^GATA3^+^ gate (51 events vs. ~5000 for other samples). A threshold of 1000 events per gate was enforced in order to ensure a representative population was obtained. Consequently this one sample was omitted from the cytokine analysis.

11) Figure 5. Please eliminate the statement on a trend (see above).

This statement has been removed.

12) Statistics: Please prove normal distribution of values or use a non-parametric test. In this regard I am particularly doubtful about Figure 5! In the same panel: If the authors wish to show that more mice with intervention than with the control cleared the infection – as elaborated in the text – a completely different analysis is required. Otherwise, please remove the statement and adjust the statistics.

We apologise for initially using the wrong statistical tests in the analysis of the pooled data represented in Figure 5. The data in Figure 5 has now been re-analysed using ANOVA with experiment and treatment as independent variables. We verified that the residuals followed a normal distribution confirming that the models assumptions were met. We tested for interactions between experiment and treatment to confirm that treatment effects were not dependent upon experimental repeat (i.e. all repeats showed the same effect). – The significant treatment effect on worm recovery, F4/80lo, monocytes and% GATA3^+^CD4^+^ expressing IL-4 were maintained with this analysis.

To give specific details for the data behind Figure 5. There was a significant effect of treatment (MC-21 versus IgG) of p<0.0212. There was also a significant experimental effect (p<0.0178). However, there was no significant interaction between experiment and treatment (p<0.3216). This indicates that the different repeats showed significantly different baseline infection levels, but that MC-21 had the same treatment effect in each repeat (interaction was non-significant). Variation in infection levels between experiments is normal. Thus, there is a significant effect of MC-21 treatment on parasite burden.

On reanalysis of the data from Figure 4, we found the data failed to meet the conditions for a parametric test, thus non-parametric Kruskal wallis test was carried out, followed by Mann-Whitney. Using this analysis, a significant difference in RELMα expression by the F4/80^lo^ population between naïve BALB/c and naïve C57BL/6 was no longer observed. Outside of this, all previously noted difference held true. Furthermore, this analysis revealed a significant difference in the expression of GATA3 between CD4^+^ T cells isolated from the pleural cavity of infected C57BL/6 and BALB/c mice. The text and figure have been adjusted to reflect these changes.

13) Figure 1: The authors state that the presented data are pooled from two separate time course experiments (day 11 + 28 and day 35 + 50). However, the data are presented as continuous line graphs, which should only be used if all data points are from one time course experiment. Please change into bar graph and explicitly label column 1 + 2 as "experiment 1" and column 3 + 4 as "experiment 2".

This has now been done.

14) It remains unclear from the text and Figure 3—figure supplement 2, why the authors chose a partial irradiation of the mice, in which not only the chest and abdomen, but also the upper extremities were protected against irradiation. Essentially, this means that the recipient mouse has quite significant residual bone marrow activity, as hematopoiesis is not restricted to long bones of the lower extremities and also take places in the upper extremities, sternum, pelvis etc. In order to calculate the contribution of bone derived-macrophages vs. resident tissue macrophages to the cellular exudates in the pleural cavity, it would have been sufficient to shield the chest alone in order to minimize bone marrow activity derived from the recipient; the donor (BM-derived) and recipient (tissue-resident) macrophages can be distinguished just by the congenic CD45 alleles.

We were restricted by the shield we had available. However, we fail to fully understand the reviewers concern. The level of shielding would not change the interpretation of the experiment. There will still be recipient bone marrow left in head and leg-irradiated mice, so instead of 30% chimerism one might get 70% chimerism. Although this may offer some additional sensitivity, the method of analysis would not change. Please note that an almost identical approach and level of shielding was used recently to identify the turnover of resident macrophages in skin draining lymph nodes [8].

As it stands, the authors need to better specify and explain the contribution of CD45.2^+^ (derived from donor bone marrow) and CD45.1^+^ macrophages (derived from residual recipient bone marrow) to the macrophage pool in the pleural space. Most importantly, the authors wanted to establish, whether the increased F4/80high population at day 35 and 50 post infection was the result of local F4/80high proliferation or of a recruitment and conversion of monocytes into the F4/80high pool (subsection “Resistant C57BL/6 mice maintain the F4/80^hi^ resMΦ population”, first paragraph). There is no clear answer to this question in this subsection, and probably also not possible due to the fact that recipient bone-marrow-derived macrophages expressing high amounts of F4/80 cannot be distinguished from F4/80high macrophages derived from the resident tissue macrophage population.

We believe there must be some misunderstanding here. The use of shielded chimeras is necessary because pleural cavity resident macrophage are radiosusceptible (question below) and so would be replaced by bone marrow derived cells. By calculating the ratio of chimerism in the blood to that in the pleural cavity, we can calculate the bone marrow contribution to the resident macrophage pool in the steady state and during infection. Because the chimerism of F4/80^hi+^ macrophages in the pleural cavity does not change on infection (low BM contribution relative to the blood), the increase in F4/80^hi^ cell numbers in the infected animals cannot be explained by enhanced cell recruitment (which would be indicated by higher levels of chimerism in cells from infected vs naïve mice), and therefore must be due to local expansion. We have added text to further explain why the shielded chimeras were necessary and included references to demonstrate that this is a well-established technique for this purpose [5,8-10]. We may have relied too much on published data rather than explaining this procedure in sufficient detail.

15) Has it ever been tested whether the resident tissue macrophages in the pleural space are radiosusceptible or radioresistant?

Yes, it was already well established that peritoneal macrophages are replaced from the bone marrow on full body irradiation [11] but Stephen Jenkins has firmly established this is also true for the pleural cavity. In fully irradiated mice the chimerism of F4/80^hi^ cells in the pleural cavity is identical to that of blood monocytes.

Literature Cited:

1) Taylor MD, LeGoff L, Harris A, Malone E, Allen JE, Maizels RM. Removal of regulatory T cell activity reverses hyporesponsiveness and leads to filarial parasite clearance in vivo. J Immunol 2005;174:4924– 33.

2) Taylor MD, Harris A, Babayan SA, Bain O, Culshaw A, Maizels RM. CTLA-4 and CD4+ CD25+ regulatory T cells inhibit protective immunity to filarial parasites in vivo. J Immunol 2007;179:4626–34.

3) Taylor MD, der Werf van N, Harris A, Graham AL, Bain O, Allen JE, et al. Early recruitment of natural CD4+ Foxp3+ Treg cells by infective larvae determines the outcome of filarial infection. Eur J Immunol 2009;39:192–206. doi:10.1002/eji.200838727.

4) der Werf van N, Redpath SA, Azuma M, Yagita H, Taylor MD. Th2 cell-intrinsic hypo-responsiveness determines susceptibility to helminth infection. PLoS Pathog 2013;9:e1003215. doi:10.1371/journal.ppat.1003215.

5) Jenkins SJ, Ruckerl D, Cook PC, Jones LH, Finkelman FD, Van Rooijen N, et al. Local Macrophage Proliferation, Rather than Recruitment from the Blood, Is a Signature of TH2 Inflammation. Science 2011;332:1284–7. doi:10.1126/science.1204351.

6) Jenkins SJ, Ruckerl D, Thomas GD, Hewitson JP, Duncan S, Brombacher F, et al. IL-4 directly signals tissue-resident macrophages to proliferate beyond homeostatic levels controlled by CSF-1. Journal of Experimental Medicine 2013;210:2477–91. doi:10.1084/jem.20121999.

7) Davies LC, Rosas M, Jenkins SJ, Liao C-T, Scurr MJ, Brombacher F, et al. Distinct bone marrow-derived and tissue-resident macrophage lineages proliferate at key stages during inflammation. Nat Commun 2013;4:1886. doi:10.1038/ncomms2877.

8) Baratin M, Simon L, Jorquera A, Ghigo C, Dembele D, Nowak J, et al. T Cell Zone Resident Macrophages Silently Dispose of Apoptotic Cells

in the Lymph Node. Immunity 2017;47:349–362.e5. doi:10.1016/j.immuni.2017.07.019.

9) Bain CC, Hawley CA, Garner H, Scott CL, Schridde A, Steers NJ, et al. Long-lived self-renewing bone marrow-derived macrophages displace embryo-derived cells to inhabit adult serous cavities. Nat Commun 2016;7:ncomms11852. doi:10.1038/ncomms11852.

10) Murphy J, Summer R, Wilson AA, Kotton DN, Fine A. The prolonged life-span of alveolar macrophages. Am J Respir Cell Mol Biol 2008;38:380–5. doi:10.1165/rcmb.2007-0224RC.

11) Merad M, Manz MG, Karsunky H, Wagers A, Peters W, Charo I, et al. Langerhans cells renew in the skin throughout life under steady-state conditions. Nat Immunol 2002;3:1135–41. doi:10.1038/ni852.